## PERSPECTIVE

### cAMP reduces action potential amplitude and conduction velocity over long axonal distance

**Fabrice Abate, Chaima Ajal and Dominique Debanne** 

*INSERM, UNIS, Aix-Marseille University, Marseille, France*

Email: dominique.debanne@univ-amu.fr

Handling Editors: Katalin Toth & Samuel Young

The peer review history is available in the Supporting information section of this article (https://doi.org/10.1113/JP287264#support-information-section).

Presynaptic neurotransmitter release at both excitatory and inhibitory synapses is highly controlled by cAMP. For example, the production of cAMP is responsible for long-term potentiation (LTP) at hippocampal mossy fibre synapses that is a result of the increased glutamate release. Indeed, cAMP shortens the coupling distance between presynaptic voltage-gated calcium channels and the release machinery (Fukaya et al., 2021), and thus elevates the efficacy of presynaptic calcium to trigger neurotransmitter release. Synaptic transmission is also under the control of the presynaptic action potential (AP) that, in contrast to the classical dogma, may vary in amplitude according to the context in which it is generated and may thus determine the output signal in an analogue-digital way (Rama et al., 2015; Zbili et al., 2020). However, the effect of cAMP on the presynaptic AP is largely unknown.

In this issue of *The Journal of Physiology*, Furukawa et al. (2025) have elegantly solved this question by showing that axon AP amplitude is reduced by cytoplasmic cAMP through the inhibition of the sodium current in the axonal compartment. Using a very distinguished combination of paired-recordings from Purkinje cells and deep cerebellar nuclei neurons maintained in culture and patch clamp recording from thin axons and presynaptic terminals (Kawaguchi & Sakaba, 2015), it is shown that, at short axonal paths, synaptic transmission is slightly enhanced, whereas at long axonal paths, synaptic transmission is reduced and delayed in time (Fig. 1). Then, Furukawa et al. (2025) show that these two effects observed at distal output synapses are a result of the reduction of the sodium current but not calcium current. Thus, for proximal synapses ($\sim$250 $\mu$m), the axonal length is too short to produce a significant effect, but not for distal synapses ($\sim$800 $\mu$m) because the inhibition of sodium channels is cumulative in the latter case. This effect does not involve the cAMP sensor, Epac, as Epac inhibitors have no

**Figure 1. cAMP reduces action potential amplitude and conduction velocity over long axonal distances**
Left: scheme of a Purkinje cell with proximal and distal synapses. At proximal synapses, the cAMP-dependent inhibition of voltage-gated sodium channel (Nav) has no effect on the amplitude of the action potential and the release of GABA is possibly augmented through the reduction of the coupling distance between voltage-gated calcium channels (Cav) and the release machinery. By contrast, at distal synapses, the cumulative inhibition of Nav channels over a long distance reduces both the amplitude and the conduction velocity, thus reducing GABA release.

effect on cAMP-induced down-regulation of axonal sodium channels. However, PKA mediates this down-regulation of sodium channels.

The study by Furukawa et al. (2025) is of great importance for several reasons. First, it shows that a single factor (i.e. cAMP) is able to differentially modulate proximal and distal outputs through the increased release at proximal synapses and the specific inhibition of axonal sodium channels that in turn reduces the amplitude of the AP and slows down its conduction at distal synapses. Such a differential effect is not common. Proximal synapses established by Purkinje cells usually correspond to local contacts on other Purkinje cells or to autaptic contacts (i.e. self-inhibitory). In other cell types such as cortical pyramidal neurons, proximal inputs locally target other pyramidal neurons. In addition, this remarkable study confirms that the fine tuning of AP amplitude has major consequences on synaptic transmission and synaptic delay. The increased synaptic delay is due to the reduction in the conduction velocity as a result of sodium channel inhibition. The study by Furukawa et al. (2025) also opens several fundamental questions. First, sodium channels are inhibited by many neuromodulators such as dopamine, serotonin or ACh through protein kinase A or protein kinase C and there are good reasons to assume that these neuromodulators may also induce similar reductions in both synaptic transmission and conduction speed at distal output synapses. In addition, the study shows that the AP amplitude is reduced in the axonal compartment but not in the somatic compartment, suggesting that the function of sodium channels is near critical in the axon because of a low density and/or a larger inactivation of axonal sodium channels compared to somatic ones. There is little doubt that future studies will address all these pending issues.

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

## Additional information

### Competing interests

No competing interests declared.

### Author contributions

All authors have approved the final version of the manuscript submitted for publication and agree to be accountable for all aspects of the work. All persons designated as authors qualify for authorship, and all those who qualify for authorship are listed.

### Funding

Agence Nationale de la Recherche (ANR): Dominique Debanne, ANR-23-CE16-0020-01; Agence Nationale de la Recherche (ANR): Dominique Debanne, ANR-21-CE16-0013-01.

### Keywords

action potential, axon, cerebellum, conduction, neuron

### Supporting information

Additional supporting information can be found online in the Supporting Information section at the end of the HTML view of the article. Supporting information files available:

**Peer Review History**

