## [Peer Review History · The Journal of Physiology]

cAMP reduces action potential amplitude and conduction velocity over long axonal distance

Dominique Debanne, Fabrice Abate, and Chaima Aja
DOI: 10.1113/JP287264

Corresponding author(s): Dominique Debanne (dominique.debanne@univ-amu.fr)

The following individual(s) involved in review of this submission have agreed to reveal their identity: Shin-ya Kawaguchi (Referee #1)

Review Timeline:

Submission Date:	15-Jul-2024
Editorial Decision:	29-Jul-2024
Revision Received:	31-Jul-2024
Accepted:	06-Aug-2024

Senior Editor: Katalin Toth

Reviewing Editor: Samuel Young

Transaction Report:

Dear Dr Debanne,

Re: JP-P-2024-287264 "cAMP reduces action potential amplitude and conduction velocity over long axonal distance" by Dominique Debanne, Fabrice Abate, and Chaima Ajal

Thank you for submitting your manuscript to The Journal of Physiology. It has been assessed by a Reviewing Editor and by 1 expert referee and we are pleased to tell you that it is acceptable for publication following satisfactory minor revision.

REVISION CHECKLIST:

We look forward to receiving your revised submission.

Yours sincerely,

Katalin Toth
Senior Editor
The Journal of Physiology

EDITOR COMMENTS

Reviewing Editor:

There are minor comments about two sentences that need additional text revision.

REFEREE COMMENTS

Referee #1:

This Perspective article is nicely highlighting the paper by Furukawa and colleagues. The text is concise and well written. I do think that the authors of original work will be deeply grateful to this Perspective which will provide the audience helpful explanation about the significance and future direction coming out of the corresponding study. I have only minor suggestions for accuracy, as listed below:

1. line 58, The Purkinje cell axon collateral will also make synapses on neighboring Purkinje cells. So, it may be better adding such words.

2. line 77, In the work by Furukawa et al., the mechanism for cAMP to augment release is not clearly demonstrated to be the shortening the coupling distance. So, adding 'possibly' before through may be better.

END OF COMMENTS

Confidential Review

15-Jul-2024

Reviewing Editor:

There are minor comments about two sentences that need additional text revision.

The two minor points have been addressed in the revised manuscript as suggested.

Reviewer #1

This Perspective article is nicely highlighting the paper by Furukawa and colleagues. The text is concise and well written. I do think that the authors of original work will be deeply grateful to this Perspective which will provide the audience helpful explanation about the significance and future direction coming out of the corresponding study.

We thank the Reviewer for his/her very positive assessment.

I have only minor suggestions for accuracy, as listed below:

1. line 58, The Purkinje cell axon collateral will also make synapses on neighboring Purkinje cells. So, it may be better adding such words.

Corrected as suggested.

2. line 77, In the work by Furukawa et al., the mechanism for cAMP to augment release is not clearly demonstrated to be the shortening the coupling distance. So, adding 'possibly' before through may be better.

Corrected as suggested.

Dear Dr Debanne,

Re: JP-P-2024-287264R1 "cAMP reduces action potential amplitude and conduction velocity over long axonal distance" by Dominique Debanne, Fabrice Abate, and Chaima Ajal

We are pleased to tell you that your paper has been accepted for publication in The Journal of Physiology.

Authors should note that it is too late at this point to offer corrections prior to proofing. Major corrections at proof stage, such as changes to figures, will be referred to the Editors for approval before they can be incorporated. Only minor changes, such as to style and consistency, should be made at proof stage. Changes that need to be made after proof stage will usually require a formal correction notice.

If you would like to receive our 'Research Roundup', a monthly newsletter highlighting the cutting-edge research published in The Physiological Society's family of journals (The Journal of Physiology, Experimental Physiology and Physiological Reports), please click this link, fill in your name and email address and select 'Research Roundup': <https://www.physoc.org/journals-and-media/membernews/>

Yours sincerely,

Katalin Toth
Senior Editor
The Journal of Physiology

P.S. - You can help your research get the attention it deserves! Check out Wiley's free Promotion Guide for best-practice recommendations for promoting your work at www.wileyauthors.com/eeo/guide. You can learn more about Wiley Editing Services which offers professional video, design, and writing services to create shareable video abstracts, infographics, conference posters, lay summaries, and research news stories for your research at www.wileyauthors.com/eeo/promotion.

IMPORTANT NOTICE ABOUT OPEN ACCESS: To assist authors whose funding agencies mandate public access to published research findings sooner than 12 months after publication, The Journal of Physiology allows authors to pay an Open Access (OA) fee to have their papers made freely available immediately on publication.

You can check if your funder or institution has a Wiley Open Access Account here: <https://authorservices.wiley.com/author-resources/Journal-Authors/licensing-and-open-access/open-access/author-compliance-tool.html>.

EDITOR COMMENTS

Reviewing Editor:

All prior criticisms have been addressed.